# Diversity of Fast-Growth Spore-Forming Microbes and Their Activity as Plant Partners

**DOI:** 10.3390/microorganisms11020232

**Published:** 2023-01-17

**Authors:** María Daniela Artigas Ramírez, Shin-ichiro Agake, Masumi Maeda, Katsuhiro Kojima, Naoko Ohkama-Ohtsu, Tadashi Yokoyama

**Affiliations:** 1Iriomote Station, Tropical Biosphere Research Center, University of the Ryukyus, 870 Uehara, Yaeyama, Taketomi, Okinawa 907-1541, Japan; 2Institute of Global Innovation Research, Tokyo University of Agriculture and Technology (TUAT), Saiwai-cho 3-5-8, Fuchu, Tokyo 183-8538, Japan; 3Institute of Agriculture, Tokyo University of Agriculture and Technology (TUAT), Harumi-cho 3-8-8, Fuchu, Tokyo 183-8538, Japan; 4Faculty of Agriculture, Tokyo University of Agriculture and Technology (TUAT), Saiwai-cho 3-5-8, Fuchu, Tokyo 183-8538, Japan; 5Faculty of Food and Agricultural Science, Fukushima University, Kanayagawa 1, Fukushima 960-1296, Japan

**Keywords:** *Bacillus*, *Viridibacillus*, *Paenibacillus*, *Trichoderma*, *Penicillium*, Fukushima, spore-forming microorganisms (SFM)

## Abstract

Biofertilizers are agricultural materials capable of reducing the usage amounts of chemical fertilizers. Spore-forming microorganisms (SFM) could be used for plant growth promotion or to improve plant health. Until now, biofertilizers based on SFM have been applied for rice and other crops. In this study, we isolated and characterized SFM, which were colonized on the *Oryza sativa* L. roots. SFM were analyzed regarding the short-term effects of biofertilization on the nursery growths. Analysis was performed without nitrogen or any inorganic fertilizer and was divided into two groups, including bacteria and fungi. SF-bacteria were dominated by the *Firmicutes* group, including species from *Viridibacillus, Lysinibacillus*, *Solibacillus*, *Paenibacillus*, *Priestia*, and mainly *Bacillus* (50%). The fungi group was classified as *Mucoromycota*, *Basidiomycota*, and mainly *Ascomycota* (80%), with a predominance of *Penicillium* and *Trichoderma* species. In plant performance in comparison with *B. pumilus* TUAT1, some bacteria and fungus isolates significantly improved the early growth of rice, based on 48 h inoculum with 10^7^ CFU mL^−1^. Furthermore, several SFM showed positive physiological responses under abiotic stress or with limited nutrients such as phosphorous (P). Moreover, the metabolic fingerprint was obtained. The biofertilizer based on SFM could significantly reduce the application of the inorganic fertilizer and improve the lodging resistances of rice, interactively enhancing better plant health and crop production.

## 1. Introduction

The nutrient supplies of plants are significantly influenced by microorganisms such as bacteria and fungi [1,2,3,4]. Those microorganisms that have a positive effect and stimulate plant growth are defined as plant growth-promoting (PGP) microorganisms [5,6,7], including those that form spores. They are distributed in the rhizosphere, on root surfaces (rhizoplane), within root tissues (root cortex), leaves, stems, and as free-living microorganisms in the soil. Currently, some mechanisms related to plant growth promotion have been clarified, such as biological nitrogen fixation, phytohormone production, nutrient solubilization, and siderophore production [5,8]. However, many factors and pathways are still unknown, especially under stress conditions.

Many environmental factors affect spore-forming microorganisms (SFM), such as spore formation and plant-interaction effectiveness. Some responses and interactions vary depending on the plant host, environmental factors, and strain origin [9]. The effect of various environmental factors on the competitiveness of PGP strains in plant hosts has been examined, but many questions are still unresolved. For example, in plant–microbe interactions, nitrogen fixation is strongly related to the physiological state of the host plant, but in other cases, such as Fe or P mobilization, most plant responses depend on the strain [1,9,10]. Therefore, a competitive and persistent SFM strain is not expected to express its total capacity for nitrogen fixation under limiting factors (e.g., salinity, unfavorable soil pH, nutrient deficiency, mineral toxicity, extreme temperatures, and soil moisture problems) [7,9]. Moreover, populations of SFM species vary in their tolerance to major environmental factors [1,10], such as what happened with the radioactive disaster in Fukushima prefecture, Japan. However, those pathways and mechanisms are poorly described. Moreover, they can indirectly affect plant growth to inhibit plant pathogenic organisms via the production of diverse anti-pathogenic chemical compounds [4,6,9].

Presently, many countries have started to develop organic farming systems for sustainable food production. As a result, some PGP rhizobacteria have been commercialized as biofertilizers. Many of these inoculants are based on strains such as *Azospirillum brasilense*, *A. lipoferum*, *A. chroococcum*, *Bacillus fimus*, *B. licheniformis*, *P. megaterium*, *Bacillus mucilaginous*, *B. pumilus*, *B. subtilis*, *Burkholderia cepacia, Deltia acidovorans*, *Paenobacillus macerans*, *Pantoea agglomerans*, *Pseudomonas aureofaciens*, *P. chlororaphis*, *P. fluorescens*, *P. solanacearum*, *P. syringae*, *Serratia entomophilia*, *Streptomyces griseoviridis*, *S. lydicus*, *Paenibacillus polymyxa*, and *Rhizobium* species [4,9].

Correspondingly, there are many published papers about the positive effects of plant growth-promoting microorganisms on various cultivated crops [2,4,10,11,12]. However, the information regarding the spore-forming microorganisms (SFM) associated with plant growth promotion is limited. The studies mainly focus on those associated with animal health [13,14], and commonly spore-forming (SF-) bacteria are more reported than fungi. Therefore, to clarify the SFM-rice interactions and mechanisms, we need a base line to understand the metabolism of these SFM [1,15]. Thus, BIOLOG is a suitable method reported in other studies as a metabolic fingerprint of some bacterial or fungal species.

In the current study, we analyzed the phylogenetic diversity and physiological characteristics of cultivated SFM from different ecological conditions as well as from rice fields as forestal rhizosphere in the Fukushima region, Japan. This study focused on the genetic diversity of fungi and bacteria that are positively associated with *Oryza sativa* L. as the main crop, which is continuously cultivated in different Japanese regions and other countries. Furthermore, we investigated the relationship between physiological characteristics as an individual and in correlation with the nutrient uptake in soils such as N, P, and K; this will lead to promising bio-inoculants for further application in crop production with low chemical fertilizer, especially in the Fukushima region. Moreover, this study will benefit the restoration of Fukushima Prefecture by helping the organic cultivation of different crops such as rice and vegetables.

## 2. Materials and Methods

### 2.1. Soil Samples and Collection Sites

The samples were collected from twenty areas in Fukushima prefecture, Japan. The samples included rhizosphere and soil (approx. 20 cm deep) (Appendix A). These areas were located in different agroecological sites with different soil types and agricultural management conditions, such as the forest area, paddy, vegetable fields, and river areas. 

### 2.2. Isolation of PGR Spore-Forming Microbes

The plant material used was *Oryza sativa* L. Two cultivars were used, Hitomebore and Koshihikari. The seeds were surface-sterilized with 70% (*v/v*) ethanol for 1 min and 2500 ppm sodium hypochlorite for 15 min and washed three times with sterile distilled water. The seeds were one last time washed with sterile distilled water for 15 min. The sterilized seeds were incubated for pre-germination for 48 h at 28 °C and incubated in darkness under sterile conditions soaked into sterile distilled water (Millipore–Sigma CA, Darmstadt, Germany). Then, the seeds were inoculated with 10 mL of rhizosphere/soil suspension solutions.

Previous to inoculation, the suspension solutions were prepared as 10 g of rhizosphere/soils in 50 mL of sterile MQ (Milli-Q water Direct-Q® 3 UV system, Millipore–Sigma CA, Darmstadt, Germany) per each suspension, and shake during 1 h at 150 rpm (Figure 1). Then transferred into 100 mL Tray-pot (58 × 45 × 54 mm; Ando-Chemical©, Nara, Japan) with 100 g of sterilized low-nutrient soil Inaho-baido (Inaho-Kako Co., Ltd., Toyama, Japan). The Inaho-baido soil was autoclaved at 121 °C/0.2 MPa for 30 min. Each sample site included 10 pre-germinated seeds. The plants were grown in a controlled environment chamber room at the Tokyo University of Agriculture and Technology (Tokyo, Japan). Plants were grown for 15 days in a photoperiod of 16 h light (5000~7000 LUX) and 8 h dark at 28 °C. A sterilized distilled water was added every day following 100% (*v/v*) of the maximum water-holding capacity of soil and was maintained at this level throughout the growth period (Figure 1).

After 15 days, the plants were harvested and washed with sterile distilled water. The whole roots were separated and crushed in 5 mL sterile MQ water until obtaining homogenate root solutions and raised up to 10 mL. Subsequently, an aliquot (50 µL) of the root homogenate solution was streaked onto Potato dextrose (PD) Agar (Difco Bacto; Becton, Dickinson and Company, Franklin Lakes, NJ, USA) and incubated for 3 days at 28 °C. The remaining solution was incubated for 1h at 65 °C, then 50 µL of root homogenate solution was transferred in Trypticase soy (TS) Agar (Difco Bacto; Becton, Dickinson and Company, Franklin Lakes, NJ, USA) and incubated 24 h at 28 °C (Figure 1). The single colonies of strains were re-streaked onto fresh plates of their respective media to obtain pure colonies. Afterward, each strain was incubated in TSB (TS-broth), and PDB (PD-broth) (Difco Bacto; Becton, Dickinson and Company, Franklin Lakes, NJ, USA), and an aliquot (1 mL) of strain-broth was kept with 1 mL of 50% glycerol stock at −80 °C for further characterization [15]. All isolates were re-inoculated onto the host for further selection and characterization.

### 2.3. Plant Assay

Three hundred twenty bacterial strains and sixty-six fungal strains were grown in PD and TS broth (respectively) between 1–3 days at 120 rpm, 28 °C to obtain 10^8^ cells mL^−1^. The cell inoculants were obtained by collecting the cell at 10,000 rpm for 5 min at 4 °C. Before inoculation, the *Oryza sativa* L. seeds were surface-sterilized, as previously mentioned.

The rice seeds were sown into 200 g of sterilized low-nutrient soil Inaho-baido (Inaho-Kako Co., Ltd., Toyama, Japan) contained in a plant box (7.6 × 7.6 × 10.2 cm). Then, five milliliters of each bacterial or fungal suspension were used to inoculate the seeds, each suspension containing 10^7^ CFU mL^−1^. The plant box contained 5 seeds, and each treatment had 3 replications. 

The plants were cultivated for 3 weeks under a controlled growth chamber room at 16 h light (5000~7000 LUX) and 8 h dark photoperiod at 28 °C. A sterilized RO water was added every day following 100% (*v/v*) of the maximum water-holding capacity of the soil. Additionally, two control were used in this step: one was non-inoculated treatment as a negative control (named control). The other was positive control, an inoculated treatment based on SF-bacteria (Biofertilizer “Yume-bio”), and it was named TUAT1(+), which contains *Bacillus pumilus* TUAT1 (AP014928.1) as rice plant growth promoting rhizobacteria (PGPR) [16]. After 3 weeks, shoot lengths and fresh mass of roots and shoots of seedlings were measured and weighed.

### 2.4. Isolation of Genomic DNA

A total of 33 strains between bacteria and fungus were selected based on their plant assay performance compared to the controls. Isolates were grown in TS (for bacteria) and PD (for fungus) broth at 28 °C for 2 days and the collected cells were washed twice with PBS buffer at a ratio of 1:1 (*v/v*). Afterward, the cells were kept together with 1 bead per sample for 1 h at −80 °C and then crushed using Tissue-lysser II (QIAGEN®, Austin, TX, USA) for 2 min. The genomic DNA was extracted using a Wizard Genomic DNA purification kit (Promega, Madison, WI, USA). The DNA concentrations were confirmed using a NanoDrop 2000 UV–vis spectrophotometer (Thermo Fisher Scientific, Wilmington, DE, USA). 

### 2.5. DNA Amplification and Sequencing

The PCR amplification and sequencing analyses were by using different housekeeping genes. For bacteria, the primer sets used for the 16S rRNA gene were carried out as described by Habibi et al. [17]. The RNA polymerase gene subunit β (*rpo*B) primers were described by Ki et al. [18]. For fungus, the primer sets used for the 18S rRNA gene were described by Wu et al. [19], as ITS-1 was carried out as described by Wang et al. [20]. Amplifications were performed using a thermal cycler GeneAmp PCR system 9700 (Applied Biosystems, Waltham, CA, USA). The PCR conditions were denaturation at 95 °C for 4 min, 30 cycles of denaturation at 94 °C for 1 min, annealing at 60 °C (ITS, *rpoB*) or 55 °C (16S and 18S rRNA) for 45 s, and extension at 72 °C for 2 min, followed by a final extension at 72 °C for 5 min [21]. The PCR products were checked using a 1.5% (*w/v*) agarose gel with 0.5 × TBE buffer mixed with 0.5 µg mL^−1^ ethidium bromide [21]. A 1 kb DNA ladder was used as a marker. Subsequently, the bands were excised for all genes, and DNA was purified using a FastGene^®^ agarose gel/PCR extraction kit (Nippon Genetics, Tokyo, Japan). According to the manufacturer’s protocols, PCR products were sequenced using an ABI Prism 3500 Genetic Analyzer (Applied Biosystems, Waltham, CA, USA). The obtained sequences were aligned using the ClustalW method (Genetix version 11), then compared in the GenBank database by using the online software BLAST algorithm-based sequence alignment (https://www.ncbi.nlm.nih.gov/genbank/, accessed on 15 January 2022). Then, the phylogenetic trees were constructed with the Maximum Composite Likelihood model (no tree-topology was used) by using the software MEGA version 12.0.

### 2.6. Physiological Activity Profile of Fukushima SFM

The selected isolates were first grown in PD and TS broth for 24 h for bacteria and 3 days for fungus at 28 °C, and 5 µL of cell suspensions at 10^7^ cells mL^−1^ were transferred onto agar and broth media and incubated for one week at 28 °C. Growth of isolates was estimated in comparison to the control treatment (non-stress) as follows: –, no growth; +, weak growth (10–20% compared to control); ++, good growth (30–60% compared to control); and +++, excellent growth (similar to/same as control) [21]. After one week, the colony-forming units (CFUs) were calculated by plate counting under stress conditions. These experiments were performed thrice for each isolate.

The isolates were evaluated for growth under different abiotic-stress conditions: alkaline (10 pH, NaOH 1M), acidic pH (4.5, HCl 5M), and high salinity (up to 4% NaCl). The ability of isolates to grow at different stress conditions was compared with the control, which was pH 6.8 and 0.1% NaCl. For this step, Yeast Mannitol Agar and TS Agar were used (Wako Pure Chemical Industries, Co., Ltd., Osaka, Japan).

### 2.7. Phosphorous (P) and Potassium (K) Solubilization Assess

Bacterial isolates were grown in the TS broth medium at 28 °C for 24 h, then 5 μL (10^7^ cells mL^−1^), and the fungus isolates were grown in the PD broth medium at 28 °C for 48 h. Each culture was spotted onto Pikovskaya’s media [22] to test P solubilization and Aleksandrow’s agar [23] to test K solubilization, and the following steps were as described by Artigas et al. [21], for fungi were incubated for 5 days, and bacterial strains were incubated 120 h, both at 28 °C in the darkness. The recording was every 24 h. The formation of a clear halo zone around the bacterial colony indicated the solubilization activity on Pikovskaya’s or Aleksandrow’s agar, and their solubilization index (SI) was calculated as [Halozone diameter + colony diameter (mm)/colony diameter (mm)] [22,23]. Experiments were performed with triplicates from the incubation of each strain. 

### 2.8. Iron (Fe) Sequestration Performance and Growth in N-Free Media

The iron sequestration and the siderophore production were performed by following the CAS assay by using CAS agar plates described by Louden et al. [24]. The isolates were grown in their respective broth medium (PDB: fungus and TSB: bacteria) at 28 °C, then 5 μL (10^7^ cells mL^−1^) of each culture was spotted onto CAS agar plates. Growth of species and CAS reaction were evaluated every 24 h up to 120 h, and compared with the control (non-iron media), all strains were non-stressed. The isolate’s growth under nitrogen-free media (N-free media) was recorded, and Ashby agar plates were used as described by Allen [25].

### 2.9. Metabolic Fingerprint of SFM from Fukushima

SFM isolates were grown in TS/PD broth medium at 28 °C for 48 h (in the vegetative stage). Subsequently, the strains were centrifuged at 4× *g* rpm at 10 °C; the pellet was kept and re-suspended in 1 mL sterile MQ water. An aliquot of 100 μL (10^7^ cells mL^−1^) of each strain suspension was spotted onto each well of the Biolog^™^ GN2 Microplates, according to the manufacturer’s protocols (BIOLOG^™^ Inc., Hayward, CA, USA). The microplates were read by using SpectracMax Paradigm Microplate reader (Molecular Devices LLC., San Jose, CA, USA), and the data Acquisition and analysis were obtained by SoftMax Pro 6.3 (Molecular Devices© LLC., San Jose, CA, USA).

Statistical analyses included Dunnett’s, Tukey’s, and other tests, which were undertaken using the EZR 1.52 software (Saitama Medical Center, Jichi Medical University, Saitama, Japan), which is a graphical user interface for R (The R Foundation for Statistical Computing, Vienna, Austria) [26]. A value of *p* ≤ 0.05 was considered an indication of statistical significance.

## 3. Results

The characteristics of the site sampling and type of source used for spore-isolation strains are shown in Appendix A. Twenty samples were taken with different vegetation histories, from forestal areas to rice fields (indicated as paddy fields). The samples included soils taken 20 cm or deeper and rhizosphere, including roots and organic matter distinguished as litter in this study. 

A total of 386 isolates were collected from the roots of both rice cultivars. From soils, 117 isolates were obtained, and 269 isolates (71% of the total) were collected from the rhizosphere (Appendix A, Figure 2a). Within bacteria or fungi, rhizosphere samples have a large number of spore-forming (SF) microorganisms (SFM) with fast growth within 24 h (for bacteria) or 72 h (for fungi). In the collected rhizosphere, 222 bacterial strains and 47 fungi were present. In contrast, there were 98 bacterial strains and 19 fungus isolates from soils (Figure 2b). Moreover, the number of isolates was predominated by SF-bacteria independently of the sample (Figure 2b). The litter samples from the forest area (Sample 4) had the highest among the sporulating microorganisms. However, fungal strains could not be collected from samples 12, 14, and 17 (Appendix A).

All SFM were re-streaked onto their correspondent media for pure colonies confirming the fast growth selected term in this study; it is important to mention that in some cases, the SFM continued to appear after keeping the plates under longer incubation. 

This study tested 320 bacterial strains and 66 fungal isolates for screening. As a host trap, these strains were re-inoculated into the ‘Koshihikari’ rice cultivar. At least two strains from each sample type showed the best plant performance confirmation between them, and in comparison with the positive control (the biofertilizer) and negative control (non-inoculation), the top representative groups with the best plant performance were selected. However, only 28% of the SFM were selected, and 18 bacterial strains were isolated from both sample types (4 from soil and 14 from rhizosphere). Furthermore, a representative group of fungal isolates was selected (15 strains, 23% of the total).

### 3.1. Phylogenetic Analysis of Fukushima Sporulating Isolates

The 16S rRNA gene and *rpoB* gene sequences of eighteen bacterial strains were used for phylogenetic analysis. In the case of fungal isolates, ITS1 and 18S rRNA gene sequences of fifteen fungal strains were used for their identification (Table 1).

On the one hand, the phylogenetic tree results of “Bacteria isolates” were similar between both genes (Appendix A), which showed that Fukushima isolates were clustered into the *Bacilliales* group. Interestingly, in this study, the SF-bacterial isolates were closely related to the *Firmicutes* phylum, only the *Bacilli* class, independently of the gene used. Then, the clustering analysis results for both genes divided the isolates into six genera groups (Appendix A) as follows: *Bacillus* (50% of total bacterial isolates)*, Priestia* (11%)*, Peanibacillus* (22%)*, Viridibacillus* (6%), *Solibacillus* (6%)*,* and *Lysinbacillus* (6%). 

*Bacillus* genus is dominant in this study, and according to the species, 9 isolates were divided into 3 sub-groups, establishing a relationship between Fukushima isolates and *B. pumilus* (4 strains), *Bacillus* sp. close to *B. thuringiensis* (3 strains) and *B. simplex* (2 strains) formerly known as *Peribacillus simplex* (Appendix A). The subsequent group was the *Paenibacillus* genus which included 4 strains and somehow closely correlated to *Paenibacillus amylolyticus*; most of the strains were classified as *Paenibacillus* sp. (Appendix A) in consensus with all genetic results (Table 1). The remaining isolates are divided into one strain for each genus (Table 1, Appendix A).

Regarding the frequency of bacterial isolates, the *Bacillus* isolates were mainly collected from soil samples with Japanese pampas grass or rice cultivation history. However, two strains, Fuk 10_10 and Fuk 4_10 were obtained from the rhizosphere. In contrast, *Priesta* isolates were only from rhizosphere samples. In the case of *Paenibacillus,* these strains were collected equally from the soil and rhizosphere. Furthermore, the *Viridibacillus* strain was collected from the rhizosphere (Appendix A), and up to date, it is the first time to describe these genera as plant growth promoting (Table 1).

On the other hand, “Fungal isolates” were genetically analyzed based on the sequences of ITS1 (Appendix A) and 18S rRNA (Appendix A) gene sequences, and their consensus species per each selected isolate is summarized in Table 1.

In this study, the fungus strains were divided into 3 big groups in the fungi taxonomy. The predominant group in this study is closely related to the *Ascomycota* division (80% of the total fungal strains), followed by the *Mucoromycota* phylum (13%) and *Basidiomycota* division (7%), independently of the phylogenetic gene used. The clustering analysis results for both genes divided *Ascomycota* division into five genera groups (Appendix A) as follows: *Trichoderma* (27% of total fungus strains), *Penicillium* (27%), *Fusarium* (13%), *Sarocladium* (7%), and *Bisporella* (7%). *Trichoderma* was mainly related to three species: *T. polysporum*, *T. lixii,* and *T. reesei* (Appendix A). In the case of *Penicillium* genus, the isolates were closely related to *P. decumbens* or *P. brasilianum* and *P. oxalicum* (Appendix A). Furthermore, the *Ascomycota* isolates were related to two species as *F. equiseti* and *F. oxysporum*. The remaining *Ascomycota* strains were closely related to *Sarocladium bactrocephalum* and *Bisporella discendens*.

In contrast, *Mucoromycota* and *Basidiomycota* isolates were closely related to one specie, respectively (Appendix A). Fuk 38 and Fuk 39 were related to *Mortierella alpina* (*Mucoromycota*), and Fuk 6 was related to *Cutaneotrichosporon moniliiforme* (formerly known as *Trichosporon moniliiforme*). Regarding the occurrence of fungal isolates, the highest number was rhizosphere samples, with 87% of the total (Table 1). Only two isolates were collected from soil; both were found in soil with Japanese pampas grass cultivation history. Furthermore, to date, it is the first time that *Cutaneotrichosporon,* and *Bisporella* isolates were described as plant growth promoting (Table 1). Additionally, several fungal strains have been described as pest biocontrol, cellulose degradation, endophytic fungus, or even a pathogen with phytohormone production activity.

### 3.2. Physiological Profile of SFM Based on Plant Growth Requirement

All genetically categorized isolates were phenotypically characterized and assessed for different abiotic stress. First, all isolates were tolerant to high-temperature, especially SF-bacteria. Salinity tolerance was determined by recording growth rates compared to the control (media without any modifications). A total of 89% of bacterial isolates and 80% of fungi could grow at high NaCl concentrations (4%) and showed good or the same growth as the control (Appendix A). However, two bacterial isolates, Fuk 11_7 (from the soil) and Fuk 13_15 (from the rhizosphere) could not tolerate NaCl at 4%; these bacterial isolates belonged to *Bacillus* and *Viridibacillus*, respectively (Table 1). Similarly, three fungal strains, Fuk 44, Fuk 49, and Fuk 36 could not grow at 4% NaCl but could grow at lower salinity (Appendix A); all of them were isolated from rhizosphere samples and belonged to the *Trichoderma* genus (Table 1).

Fukushima SFM showed tolerance to different pH conditions, from acidic to alkaline. Surprisingly, fungal isolates were not affected by pH changes. However, bacterial isolates were more affected by acidic than alkaline pH. The isolates Fuk 11_7 and Fuk 11_10 (from rhizosphere) could not grow under acidic conditions. Moreover, Fuk 11_7 could not grow at 10 pH. Figure 3(a1,b1) showed the relation between the stress abiotic and the SFM strains, indicating that one strain is out in the SF-bacteria strains, which is identified as Fuk 11_7 (*Bacillus* sp.); it is a very sensitive bacterial strain to any abiotic stress. NaCl only affects the growth of fungus (Figure 3(b1), Appendix A).

The isolates growth was accessed based on nutrient assimilation as plant partners such as P or K solubilization, Fe chelation, and capacity of growth in N-free media (Appendix A). The correlation results between nutrient and isolates are shown in the Venn diagram (Figure 3(a2,b2)). In the case of bacterial isolates, most of the isolates could grow under different media with inorganic K and P source, N-free media, but no isolate could grow and assimilate Fe in the media used in this study (Figure 3(a2), Fe is absent in the group). For K and P, solubilization results showed that almost all isolates (89%) could grow under these conditions. The K solubilization ratio is low, and no significant difference was found between strains. In contrast, P solubilization in the isolates showed significate differences, and six isolates (33% of the total) could grow but did not show solubilization. Those isolates with high P solubilizing index were 56%; these ten isolates also shared a positive correlation with another tested nutrient (Figure 3(a2)). In ascending order of isolates’ solubilization index for P as follows: Fuk 4_8, Fuk 4_31, Fuk 10_6 and Fuk 10_10, Fuk 5_23 and Fuk 13_14 with P index of 1.0, Fuk 11_10 was 1.2, Fuk 7_4 and Fuk 8_6 were 1.3, and the highest P index was the isolate Fuk 11-2 (*Peribacillus*; Table 1 and Appendix A). In the case of N, most of the isolates (94%) showed N_2_ fixation (in Ashby media). Similar to previous stress results, the Fuk 11_7 (*Bacillus* sp.) could not grow with inorganic K or P and under N-free media. 

On the other hand, fungus isolates could not grow in media supplemented with inorganic K (Figure 3(b2), K is the out-group). For Fe chelation in the vegetative stage, only 3 isolates could grow (20%) and showed a good Fe index at 2.0 as Fuk 58 (*Fusarium*), Fuk 10, and Fuk 66 (*Penicillium*). In the case of P solubilization, 93% could grow in media supplemented with inorganic P, but only 20% (3 isolates) showed good solubilization. These isolates were Fuk 6 (*Cutaneotrichosporon* with 1.0 index), Fuk 50 (*Penicillium* with 1.1 value), and Fuk 59 (*Bisporella* with 1.3 value) with a significative P solubilization index. Only Fuk 49 (*Trichoderma*) could not grow in media supplemented with inorganic P. N_2-_fixation (in media) showed in 80% of the isolates; Fuk 36, Fuk 40 (*Trichoderma*) and Fuk 63 (*Penicillium*) could not grow in N-free media. 

### 3.3. Metabolic Profile of Fukushima Isolates at the Vegetative Stage

Biolog^TM^ microplates were used in this study as phenotyping and metabolic fingerprint on the Fukushima spore-forming microorganisms. The results showed some correlation between some substrate and phylogenetic clustered groups. Firstly, SFM isolates in this study are correlated for being incapable of metabolizing or assimilating D-psicose and L-arabinose; those that showed some kind of reaction were inconclusive (Figure 4: SF-bacteria, Figure 5: Fungi, Appendix A).

In this order, the SF-bacterial isolates as shown in Figure 4. Bromosuccinic acid, i-erythritol, and β-methyl-D-glucoside were assimilated in almost all strains isolated in this study, except for Fuk 10_10 and Fuk 5_22, with inconclusive results, which are in the borderline according to the Biolog^™^ protocols.

In detail, the *Priestia* group is capable of metabolizing 77 compounds; however, these isolates could not grow or use the following sources such as α-D-glucose, D-mannose, and the organic acids such as formic, γ-Hydroxybutyric, and glutamic acids. While the isolates related to *B. pumilus* could metabolize acetic and citric acid, but they could not grow with D-galactose as the only carbon source. In the remaining compounds, the response varied depending on the isolate. The *B. thuringiensis* group shares certain characteristics with each strain as incapable of metabolizing α-D-glucose and D-gluconic and D-saccharic acids. These strains could metabolize 18 compounds, such as gentiobiose, raffinose, and some acids, such as glucosaminic, α-ketoglutaric, and malonic acids. *Peribacillus* isolates can synthesize around 26 compounds such as glycerol-phosphates, citric and p-hydroxyphenlyacetic acids. However, these isolates cannot grow in media with carbon sources such as D-galactose, D-mannose, D-glucuronic and succinic acids.

Furthermore, *Viridibacillus*, *Lysinibacillus*, and *Solibacillus* isolates could not grow with glucosamine-bound compounds, D-mannose, and some organic acids such as malonic and D-saccharic acids (Figure 4, Appendix A). Lastly, *Paenibacillus* share the characteristics of metabolizing some organic acids, maltose, and β-methyl-D-glucoside. However, *Paenibacillus* isolates cannot metabolize 18 compounds such as organic acids such as succinic acid, and sugars such as D-galactose, and D-mannose (Figure 4, Appendix A).

On the other hand, the fungal isolates’ metabolic profiles were examined, as shown in Figure 5. These fungal isolates were correlated by the positive responses to metabolize or assimilate dextrin, glycogen, D-melibiose, cis-aconitic, α-hydroxybutyric, α-ketobutyric, succinamic acids, and L-ornithine. In the case of malonic acid, lactic, acetic, propionic, formic, and α-ketoglutaric acids were compounds assimilated or metabolized by almost all fungal isolates except Fuk 59.

The fungal isolates share certain characteristics by the phylogenetic group as follows: *Trichoderma* isolates metabolize 57 compounds (Figure 5). However, these isolates cannot grow in D-cellobiose, L-fucose, and D-galactose. *Penicillium* isolates have in common that they could metabolize 34 compounds, but these isolates are not capable of growing in the presence of D-cellobiose, D-galactose, α-D-glucose, and L-rhamnose. The isolates’ responses in front of the remaining compounds depend on the strains, including inconclusive ones (Figure 5, Appendix A).

*Fusarium* could not metabolize α-D-glucose and D-glucuronic acid; however, the isolates could metabolize the remaining compounds. *Mortierella* could not grow with 23 compounds such as using α-D-glucose and D-mannitol as thesole carbon source, and organic acids such as D-saccharic, succinic, L-glutamic acids, and compounds with glucose-phosphate (Figure 5). *Sarocladium* could metabolize 65 compounds, but there were 23 compounds where there was no response with D-fructose, α-D-glucose, D-mannitol, sucrose, L-asparagine, hydroxy-L proline, and L-pyroglutamic acid. *Bisporella* isolates seem to be sensitive to any media source, and it is distinguished by metabolized adonitol, D-fructose, L-fucose, α-D-lactose, and the organic acids such as pyruvic, citric, D-galactonic, lactone glucosaminic, urocanic acids. *Cutaneotrichosporon* isolate could not grow in D-galactose, α-D-glucose, D-mannose, and succinic acid (Figure 5, Appendix A).

### 3.4. Plant Growth-Promoting Performance of SFM from Fukushima

The selected SFM from Fukushima were used as inoculants during the vegetative phase. Almost all selected isolates promoted plant growth except for two, Fuk 7_4 (*Paenibacillus* sp.) and Fuk 38 (*Mortierella* sp.), in comparison with the controls as shown in Figure 6 (SF-bacteria) and Figure 7 (Fungi). The length is positively affected in the case of the bacteria inoculants, which showed rice plant growth of >12 cm, equal to TUAT1 (positive control) or higher than the negative control (“Control: non-inoculation”, Figure 6a). In contrast, the fungus inoculants did not show any improvement in the length of the rice plants; these results are not significantly different from the negative control (Control, Figure 7a). However, we could not find any correlation between the length and biomass promotion.

The total biomass showed significant biomass with more than 85 ± 0.9 mg per nursery plant; in the case of bacteria, one isolate, Fuk 8_6 (*B. pumilus*) showed the highest promotion with 148.2 ± 0.1 mg per plant (Figure 6b). Other isolates related to *B. pumilus* (Fuk 11_1, Fuk 11_7), *Peribacillus* (Fuk 11_2, Fuk 5_23), and *Viridibacillus* (Fuk 13_15) showed significant promotion in comparison with the TUAT1 (positive control which is a commercial biofertilizer based on spore-forming *Bacillus pumilus*) with more than 103.1 ± 0.5 mg per plant. On the other hand, fungal inoculants promoted the biomass with more than 100.3 ± 0.1 mg per plant, those isolates with significate biomass promotion were classified as *Penicillium* (Fuk 66, Fuk 63) and Fuk 36 (*Trichoderma*) with more than 160 mg per plant; subsequently by Fuk 58 (*Fusarium*) and *Mortierella* (Fuk 39) with more than 150 mg per plant (Figure 7b). The remaining isolates showed high biomass compared with the TUAT1 inoculant, except for Fuk 38 (*Mortierella*) and Fuk 6 (*Cutaneotrichosporon*).

As mentioned before, biomass has been improved by SFMs as inoculants. However, the best performance in each plant group in some cases differs, such as Fuk 5-23, which showed the highest shoot mass with 76.5 ± 0.2 mg (Figure 6c); but in the case of fungi, there is no higher promotion than TUAT1 (Figure 7c). In the root case, two bacterial isolates (Fuk 11_7: 66.5 ± 0.3 mg and Fuk 11_1: 61.2 ± 0.2 mg) showed significant difference compared with TUAT1 (45.8 ± 0.1 mg), as shown in Figure 6d. However, the fungal inoculants showed great improvement in root mass and root hairs, being significantly positive and higher with Fuk 6 (77.9 ± 0.2 mg), Fuk 9 (74.5 ± 0.3) mg, and Fuk 16 (73.7 ± 0.1 mg) (Figure 7d).

In summary, according to the analysis of the top rice promoters at the seedling stage, the SF-bacterial isolates were Fuk 8_6 (*B. pumilus*), Fuk 11_7 (*P. megaterium*) and Fuk 13_15 (*Viridibacillus* sp.); the first two strains were isolated from soil and the last one from the rhizosphere. For fungus isolates, they were Fuk 66 (*Penicillium* sp.), Fuk 63 (*Penicillium* sp.), and Fuk 36 (*Trichoderma*); these strains were mainly isolated from the rhizosphere. Interestingly, we correlated that these SF isolates have the ability or the metabolic pathway to work under inorganic P corroborated by media and by assimilation of phosphates compounds.

## 4. Discussion

The diversity of SFM in spore form are dormant species with great importance worldwide and have been examined with various techniques and criteria [14,27,28]. However, most of the descriptions are focused on public health and poorly describe them from the perspective of plant–microbe interactions or as an inoculant [4,14]. Our study focused on fast-growing SFM isolation and their characteristics as future plant partners with fast break dormancy strains, and their positive effects on the plant growth or health. This is because a slow-growing group of microbes could not compete with fast-growing bacteria and fungi, especially during the early phases of crop development where microbial diversity establishes itself, even after harvest [3,4]. Additionally, our study will increase the knowledge of SFM in Fukushima prefecture after a long-term halt to crop cultivation in many sampling areas.

The SFM were found independently from rhizosphere and soil samples due to the large range of environmental conditions [27,28,29] in Fukushima prefecture. However, low diversity of SFM or absence was observed in some samples, which indicates that the method used may not be the most suitable for studying SFM diversity. Other reasons, such as the selection method, also need a longer period for isolation (≥48 h), and out of the isolates reported in this study other species may need to take more than 72 h to break spore dormancy. Lastly, some SFM might be uncultivable because the media use is unsuitable or the strains become uncultivable after changing their ecological niche [13,28,29,30]. Our results indicate that the rhizosphere is a better source than the soil and bacterial isolates are more frequent and faster growers than the fungal type [3,13]. This is attributable to the ability of these SF-bacteria to better adapt than fungi, which gives them the capability to survive abiotic stresses, nutrient deficient, and compound productions such as biofilms [27,31].

In this study, we tried to cover the fast-growing SFM independently from the season due to decomposition deceleration; bacterial and fungal biomass and sporulation rates change according to the type of origin sample or area in one region whether it is a wetland, temperate stream, or arable land, in concordance to previous reports [13,32,33,34,35,36]. SFM populations with different ecological or agricultural conditions are active colonizers of these ecosystems, thereby showing some specialized physiological functions associated. It was also observed that in the field with different stages of plant succession or rice development, bacteria and fungi seem to be the main colonizers, which was in agreement with our results wherein the bacteria were dominant [3,28,29,34,35,36,37,38,39,40,41,42].

Similarly, our results suggested that the class *Bacilli* dominated fast growth and fast-breaking spore dormancy [34]. Garbeva et al. [34] described SF-bacteria as *Bacilli*-related community structures with high diversity among the order *Bacillales*, including *Bacillus* as main genus, their changes have been found under different agricultural regimes such as in permanent grassland, arable land, and under long term presence of grass in soil [34,35,36]. In contrast, our results are somewhat similar but include the predominance of filamentous fungi (*Ascomycota*) as the main group, which is supported by Sharma et al. [3]; it described that in paddy fields under different growth or decomposition stages, the fungal community and their ratio changes, and not only SF-bacteria [15]. Thus, these results could be associated more with the significant interaction between the time of decomposition and emersion for sporulation of fungal communities, which is related to the enzymatic process of C mineralization, N, P, and S cycles [3,28,29,33,34,35].

Furthermore, SFM development is commonly attributed to the level of adaptation to a specific niche in terms of the ability to utilize available nutrient resources and tolerate metabolite accumulation or deficiency, such as proline, glycine, trehalose, sucrose and mannitol; on the other hand, these compounds cause oxidative and osmotic stresses in plants [9,15]. Our results indicated that SFM isolates utilize or metabolize these compounds, as demonstrated by Biolog^™^, and indirectly suggested that the presence of SFM in the soil might improve the rice seedling under stress or stress tolerance in plants. Moreover, the Biolog^™^ method was used as an Eco-Plates, which can provide an overview and screening of the metabolic fingerprint characteristics of SFM in a short time [31,34,38,39].

The physiological profiling is accompanied by microbial identification for future investigations of microbial community structure and applications in the field [13,30,37,38]. The immediate results can be compared with other studies; for example, until the 2020 outcome, pathogenic spore-formers do not grow below pH 4.5 and the species encountered in spoilaged foods are not the same as in our study [14,27,39]. Therefore, the metabolic profile and other physiological results supported the adequate selection of SFM and the isolates’ ability for use as a possible response to climatic changes or different crop cultivation due to its adaptability [13]. Moreover, our results suggested that SFM positively promoted plant growth through various mechanisms, such as plant hormone and ESP production, regulating nutrient exchange, and releasing stress-protectant compounds. At the same time, our SFM metabolic results support, for example, cytokinin production [1,16] by the assimilation of amino acids or sugar-bound and phosphate-bound compounds [15]. The stress-protectant compounds could be the result of SFM biosynthesis of total soluble sugar, proline, and trehalose or by regulating metabolites such as pyruvic acid, succinic acid, and uridine [9,15]. Future experimentation is needed for different SFM strains, which will help elucidate the underlying mechanisms and ecology, such as signaling, colonization, and SFM community interactions.

On the other hand, the spore resistance of SFM is related to the sporulation performance at optimal vegetative growth conditions resulting in the highest sporulation efficiency and spore resistance [14]. Until now, the major influence of sporulation was well reported to be the temperature conditions; based on this, knowing other factors could help increase their resistance to many abiotic stress factors and improve the predictability of spore behavior [1,8,14]. Additionally, this can change according to the succession in the environmental conditions and plant stage, similar to the ripening stages of SF-bacteria in cheese manufacturing [28,37].

Consequently, our results indicate that *Bacilli* isolates from Fukushima can tolerate low temperatures and acidic conditions and can also be heat-stable spores, as reported previously [1,4]. Moreover, the SFM isolates are mainly aerobically forming with a mesophilic preference (optimal growth temperature between 10 °C and 30 °C) with stable spores forming at 65 °C. Hence, previous studies indicated that endophytic *Penicillium*, *Aspergillus*, *Trichoderma*, *Bacillus,* and *Paenibacillus* reduced pathogenic fungi and also induced the biomass yield of cultivated plants [12,28,29,36,37,40] which is similar to our results in the promotion of growth for *Oryza sativa* L.; however, further experiments are required for assertive use as a future pathogenic antagonist.

Finally, physiological results and their interaction with rice can be correlated; for example, those SFM isolates capable of metabolizing D,L,α-glycerol-phosphate, α-D-glucose-1-phosphate or D-glucose-6-phosphate, also have good P abilities reported in the Pikoskaya medium. At the same time, these results work as guidelines for other unknown strains, such as *Viridibacillus*, *Lysinibacillus*, and *Bisporella* reported in this study. Those strains are reported as PGPR or are known to be P solubilizers such as *P. megaterium* [4]. Altogether, this suggests that some species could solubilize P, K, and other minerals and fix nitrogen to promote plant growth. Similarly, it could control plant diseases [3,4,13,41], especially in Fukushima area, with the relationship between K/Cs uptake. Thus, their physiological role was one factor behind their plant association or interactions throughout [1,42]. In summary, these results suggest that SF-bacteria development and sporulation may be more resilient in shorter growth periods, and their effect as inoculants may be shorter than fungi; however, the effect of the fungi could remain longer than bacteria [13].

## 5. Conclusions

The SFMs from Fukushima prefecture selected were Fuk 8_6 (*B. pumilus*), Fuk 11_7 (*P. megaterium*), Fuk 13_15 (*Viridibacillus* sp.), Fuk 66 (*Penicillium* sp.), Fuk 63 (*Penicillium* sp.), and Fuk 36 (*Trichoderma*); these SF isolates could improve plant biomass under limited nutrients and showed good abilities under different abiotic stress and nutrients restriction. Moreover, further research is necessary to confirm the effectiveness of strains for rice and other crops in field conditions with low nutrient supply as a biofertilizer or against pathogens as biocontrol.

## Figures and Tables

**Figure 1 microorganisms-11-00232-f001:**
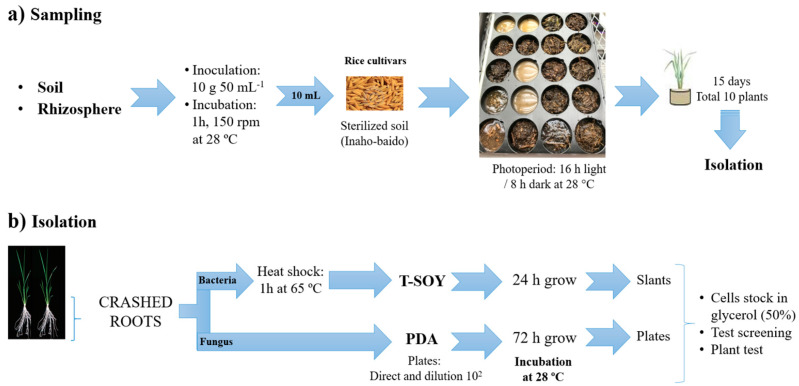
Spore-forming microbes (SFM) isolation from Fukushima prefecture, Japan. (**a**) Inoculation with rhizosphere and soil into *Oryza sativa* L. (**b**) SFM isolation from rice roots. (T-soy: Trypticase soy agar. PDA: Potato dextrose agar).

**Figure 2 microorganisms-11-00232-f002:**
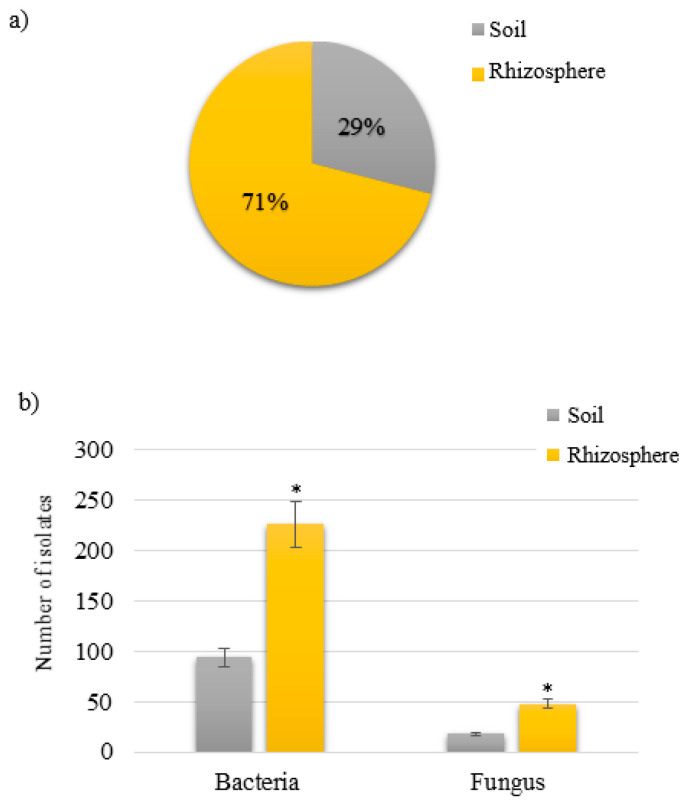
Spore-forming microbes (SFM) distribution according to the sample type. (**a**) Total percentage of SFM. (**b**) Total Number SFM based on sample type. * means a value with *p* ≤ 0.05 was considered an indication of statistical significance.

**Figure 3 microorganisms-11-00232-f003:**
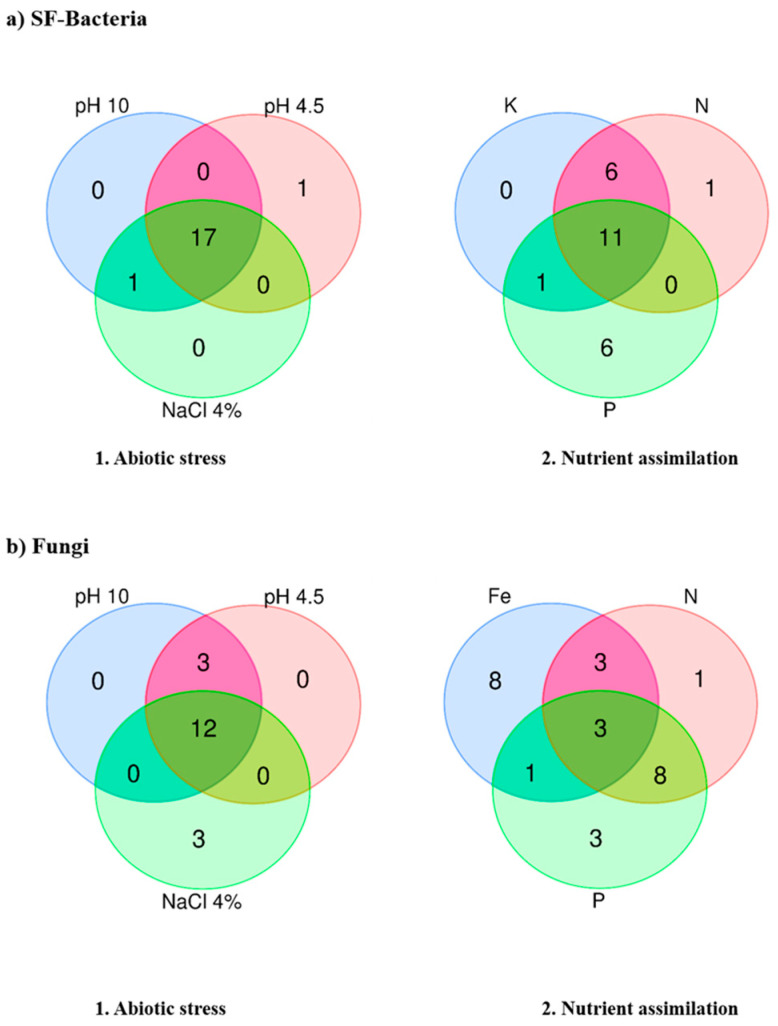
Venn diagram analysis based on abiotic stress tolerance and limited nutrients in media. (**a**) SF-bacteria isolates results based on the survival number. (**b**) Fungi isolates results based on the survival numb.

**Figure 4 microorganisms-11-00232-f004:**
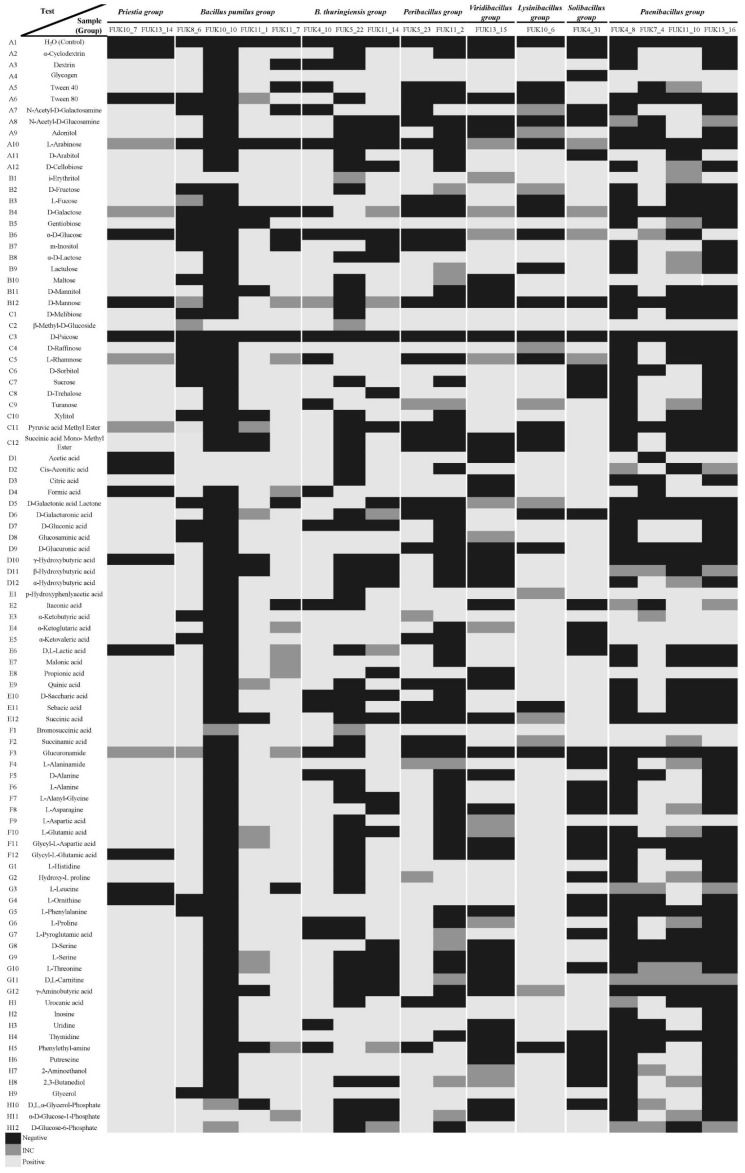
Mapping of metabolic fingerprint of SF-bacteria from Fukushima. It was obtained by excel commanding from raw reads after 24 h, 48 h, and 72 h incubation. ■ indicates those results were negative, ■ indicate those inconclusive results (INC), and ■ indicates the positive reaction from SFM isolates.

**Figure 5 microorganisms-11-00232-f005:**
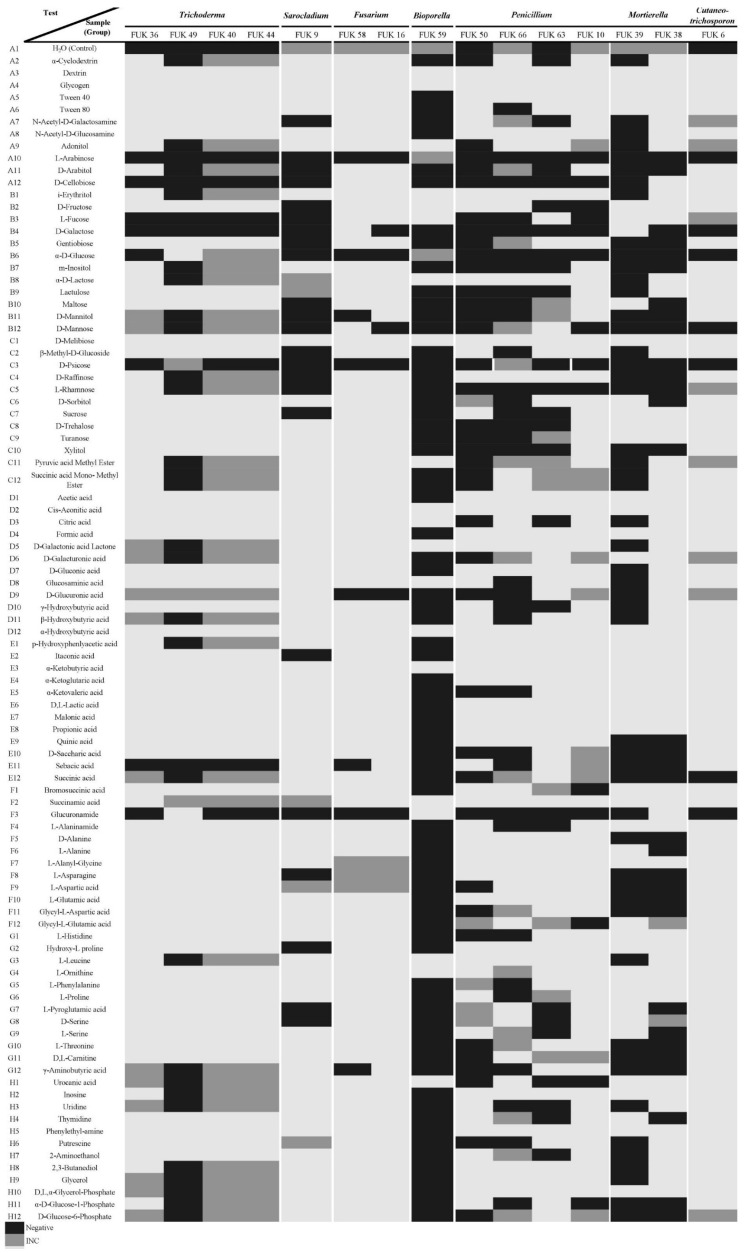
Mapping of metabolic fingerprint of Fungi from Fukushima. It was obtained by excel commanding from raw reads after 24 h, 48 h and 72 h incubation. ■ indicates those results were negative, ■ indicates those inconclusive results (INC), and ■ indicates the positive reaction from SFM isolates.

**Figure 6 microorganisms-11-00232-f006:**
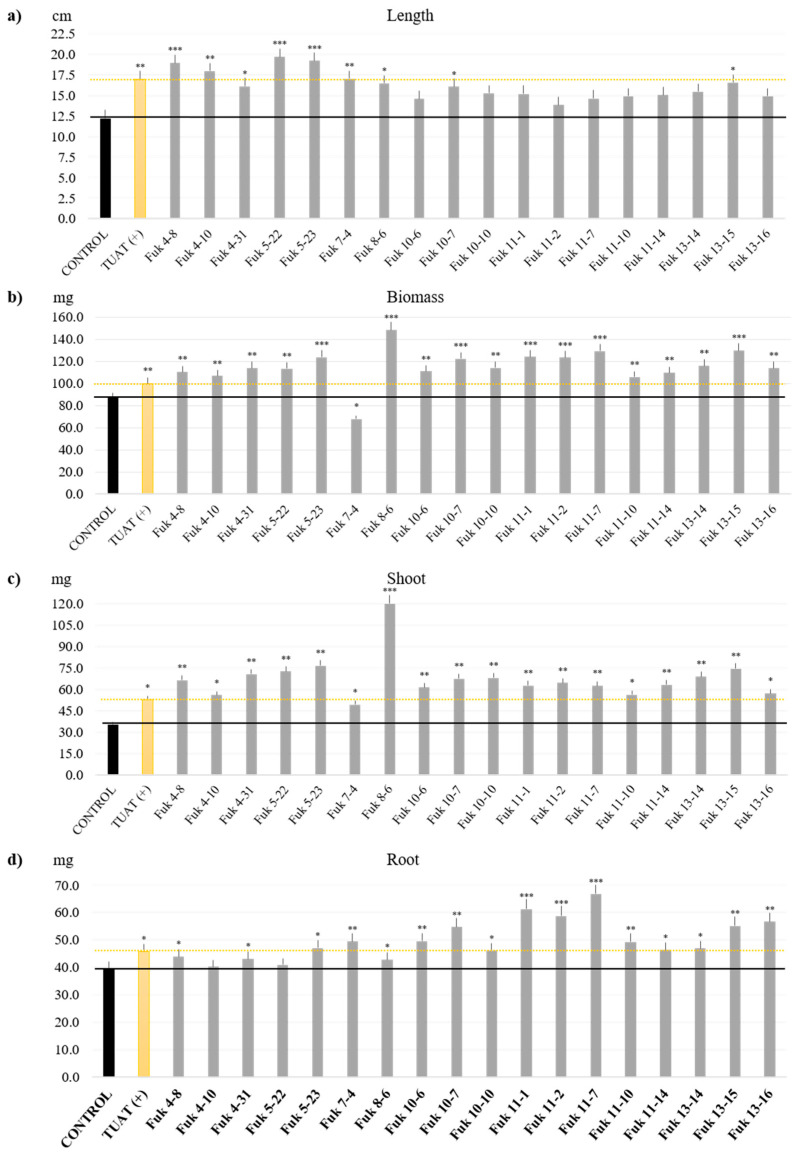
Inoculation of *Oryza sativa* L. Koshihikai with each pre-selected SF-bacterial isolates. The results are based on the total mass and length average from 5 plants with three replications for each treatment. (**a**) length (cm), (**b**) total fresh biomass (mg), (**c**) total fresh shoot mass (mg), and (**d**) total root mass (mg). Control means non-inoculation, and TUAT1(+) means positive control from based spore-forming biofertilizer. Dunnet’s test was used for length with control as based comparative, and Tukey’s test was applied for other parameters *, **, and *** means a value considered statistical significance with *p* ≤ 0.05, *p* ≤ 0.01, and *p* ≤ 0.001, respectively. No annotation means no significance.

**Figure 7 microorganisms-11-00232-f007:**
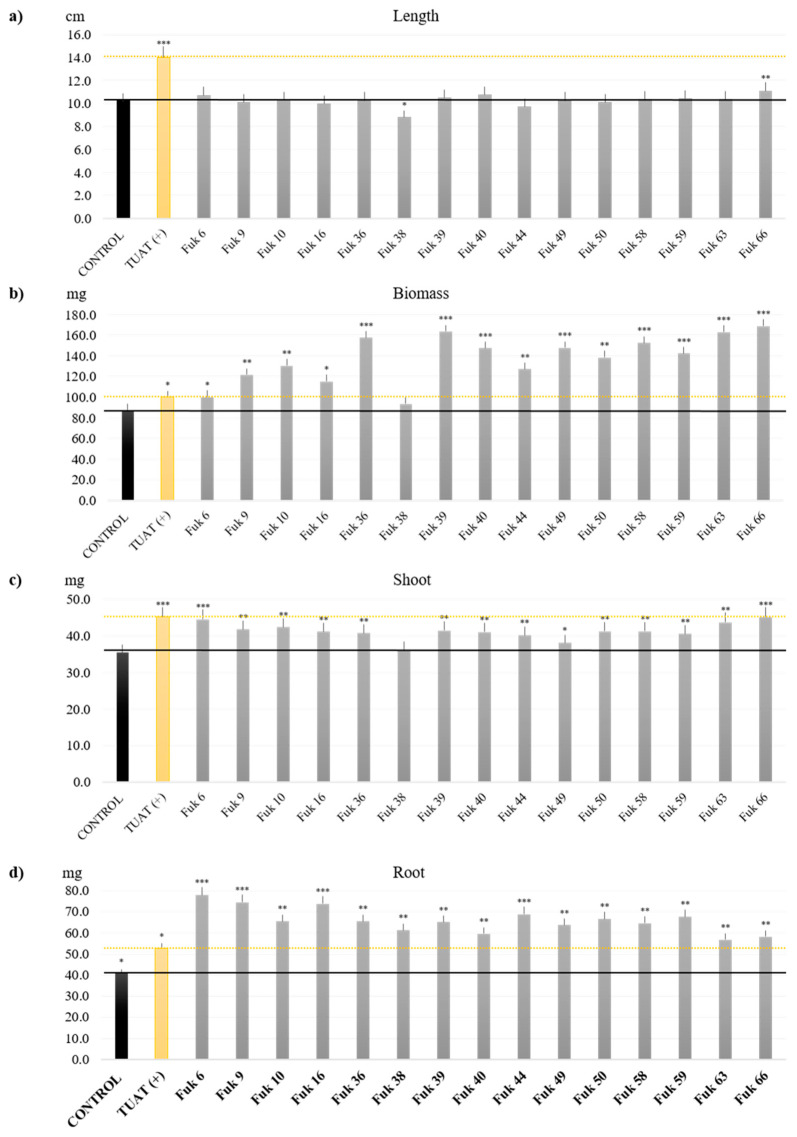
Inoculation of *Oryza sativa* L. Koshihikai with each pre-selected fungi isolates. The results are based on the total mass and length average from 5 plants with 3 replications for each treatment. (**a**) length (cm), (**b**) total fresh biomass (mg), (**c**) total fresh shoot mass (mg), and (**d**) total root mass (mg). Control means non-inoculation, and TUAT1(+) means positive control from based spore-forming biofertilizer. Dunnet’s test was used for length and control as based comparative, and Tukey’s test was applied for other parameters *, **, and *** means a value considered statistical significance with *p* ≤ 0.05, *p* ≤ 0.01, and *p* ≤ 0.001, respectively. No annotation means no significance.

**Table 1 microorganisms-11-00232-t001:** Summary of the phylogenetic results of Fukushima SF-isolates.

Group	Sample Name		Origin	18S/16S rRNA	Accession No.	*ITS/rpoB*	Accession No.	Identity (%) ^a^	Plant Interactions ^b^
**Fungus**	Fuk	6	Rhizosphere	*C. moniliiforme*	LC606568	*C. moniliiforme*	LC606569	99.80	Unknown
	Fuk	9	Rhizosphere	*Sarocladium* sp.	LC606567	*S. bactrocephalum*	LC606570	99.00	Unknown/pathogen
	Fuk	10	Rhizosphere	*Penicillium* sp.	LC606566	*Penicillium* sp.	LC606571	98.50	Unknown
	Fuk	16	Rhizosphere	*Fusarium* sp.	LC606565	*Fusarium* sp.	LC606572	98.50	Biocontrol
	Fuk	36	Rhizosphere	*Trichoderma* sp.	LC606564	*T. polysporum*	LC606573	98.00	Unknown
	Fuk	38	Rhizosphere	*Mortierella* sp.	LC606563	*M. alpina*	LC606574	98.00	Growth in roots
	Fuk	39	Soil	*Mortierella alpina*	LC606562	*M. alpina*	LC606575	100.00	Growth in roots
	Fuk	40	Soil	*Trichoderma reesei*	LC606561	*Trichoderma* sp.	LC606576	98.75	Cellulose degradation
	Fuk	44	Rhizosphere	*Trichoderma* sp.	LC606560	*T. polysporum*	LC606577	98.50	Unknown
	Fuk	49	Rhizosphere	*Trichoderma lixii*	LC606559	*Trichoderma* sp.	LC606578	99.05	Biocontrol of pest
	Fuk	50	Rhizosphere	*Penicillium brasilianum*	LC606558	*Penicillium* sp.	LC606583	94.00	Pathogen/Unkown
	Fuk	58	Rhizosphere	*Fusarium* sp.	LC606557	*Fusarium equiseti*	LC606579	99.00	Pathogen/Unkown
	Fuk	59	Rhizosphere	*Bisporella discedens*	LC606556	*B. discedens*	LC606582	98.05	Unknown
	Fuk	63	Rhizosphere	*Penicillium* sp.	LC606555	*Penicillium* sp.	LC606581	99.25	Unknown
	Fuk	66	Rhizosphere	*Penicillium* sp.	LC606554	*Penicillium* sp.	LC606580	98.00	Pathogen/Unkown
**Bacteria**	Fuk	4_8	Rhizosphere	*Paenibacillus* sp.	LC606518	*Paenibacillus* sp.	LC606536	99.00	Unknown
	Fuk	4_10	Rhizosphere	*Bacillus* sp.	LC606519	*Bacillus* sp.	LC606537	99.85	Unknown
	Fuk	4_31	Rhizosphere	*Solibacillus* sp.	LC606520	*Solibacillus* sp.	LC606538	94.00	PGPR
	Fuk	5_22	Soil	*Bacillus* sp.	LC606521	*Bacillus* sp.	LC606539	99.00	Unknown
	Fuk	5_23	Soil	*Peribacillus simplex*	LC606522	*Peribacillus* sp.	LC606540	94.20	PGPR
	Fuk	7_4	Soil	*Paenibacillus* sp.	LC606523	*Paenibacillus* sp.	LC606541	99.95	Unknown
	Fuk	8_6	Soil	*Bacillus pumilus*	LC606524	*Bacillus pumilus*	LC606542	97.00	PGPR
	Fuk	10_6	Rhizosphere	*L. fusiformis*	LC606525	*Lysinibacillus* sp.	LC606543	90.00	Biocontrol
	Fuk	10_7	Rhizosphere	*Priestia megaterium*	LC606526	*P. megaterium*	LC606544	99.00	Unknown
	Fuk	10_10	Rhizosphere	*Bacillus pumilus*	LC606535	*Bacillus pumilus*	LC606553	98.00	PGPR
	Fuk	11_1	Soil	*Bacillus pumilus*	LC606527	*Bacillus pumilus*	LC606545	99.50	PGPR
	Fuk	11_2	Soil	*Peribacillus* sp.	LC606528	*Peribacillus sp*	LC606546	98.00	PGPR
	Fuk	11_7	Soil	*Bacillus* sp.	LC606529	*Bacillus* sp.	LC606547	97.00	Unknown
	Fuk	11_10	Soil	*P. amylolyticus*	LC606530	*Paenibacillus* sp.	LC606548	99.00	Unknown
	Fuk	11_14	Soil	*Bacillus* sp.	LC606531	*Bacillus* sp.	LC606549	96.20	Unknown
	Fuk	13_14	Rhizosphere	*Priestia megaterium*	LC606532	*P. megaterium*	LC606550	99.80	PGPR
	Fuk	13_15	Rhizosphere	*Viridibacillus arenosi*	LC606533	*Viridibacillus* sp.	LC606551	98.00	Unknown
	Fuk	13_16	Rhizosphere	*Paenibacillus* sp.	LC606534	*Paenibacillus* sp.	LC606552	98.00	Unknown

Fuk: Fukushima. ^a^: Consensus identity including the phylogenetic analysis. ^b^: Based on the literature up to 2021, unknown means no reference was found about this strain positively or negatively. Highlighted means no information was found about any species of this genus.

## Data Availability

GenBank (http://www.ncbi.nlm.nih.gov/genbank, accessed on 15 January 2022). Accession Numbers: the obtained sequences for the different genes found in this study have been deposited in the DNA Databank of Japan (DDBJ) under accession numbers as follows LC606518-LC606535 for 16S rRNA sequences, LC606536-LC606553 for the *rpoB* gene, LC606568-LC606554 for the 18S rRNA gene, and LC606569-LC606580 for the ITS-1 region.

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
