# Peer review of "Diversity of Fast-Growth Spore-Forming Microbes and Their Activity as Plant Partners"

_microorganisms, 2023, doi:10.3390/microorganisms11020232_

Round 1

Reviewer 1 Report

The manuscript entitled “Diversity of Fast-growth Spore-forming Microbes and their Activity as Plant Partners'', studies how the genetic diversity of fungi and bacteria that are present associated with Oryza sativa L., cultivated in different Japanese regions. The authors related the biofertilizer based on Spore-forming microorganisms could significantly reduce the application of the inorganic fertilizer and improve the lodging resistances of rice, interactively enhancing better plant health and crop production.

The benefits of Spore-forming microorganisms in promoting rice growth opened the possibility to use these bacteria and fungi as a rice inoculant. A new perspective on the study and exploration of this biotechnological potential is the result from the identification of this bacterium and fungi, which allowed a better understanding of the role played by these microorganisms in the rice interaction.

The study presents an unprecedented panorama of how the microbiome in rice is behaving in a certain area of Japan, an area that suffers a serious environmental accident.

The methodological procedures were well executed as described, with the exception of the chemical conditions (DNA concentration, primers ....) and physical conditions (detailed description of the thermocycling cycles) in the item DNA amplification and sequencing.

The conclusions presented are consistent with the initial proposal of the manuscript, being well presented and discussed.

Cited references are appropriate for the article, most are up-to-date and from articles published in high impact factor journals. Therefore, providing substantial support to the discussion proposed in the article.

The manuscript so submitted could be accepted for publication in the journal in a Microorganisms journal.

Author Response

Thank you for reviewing our manuscript. Regarding the comments and recommendations, the manuscript was revised accordingly to your recommendations to some extend, please see the new version of the manuscript.

Reviewer 2 Report

There are lots of errors in the writing part. The language is not up to the standard and is difficult to understand. Most importantly it is a routine work of isolation, evaluation, and identification. Atleast the writing part should be improved.

Author Response

Thank you for reviewing our manuscript. Regarding the comments and recommendations, the manuscript was revised accordingly to your recommendations.

Reviewer 3 Report

Dear authors,

The manuscript presents a work analyzing the Spore forming microorganisms isolated from the soil of the Fukusima region (Japan).

In my opinion, the work needs to be corrected and improved in some points, clarifying some methodological aspects.

- Introduction should be developed and explain deeply the mecanisms of PGP bacteria.

- The sentence of the line 57 to 59 is not consistent with the list of commercial biofertilizers. Seven strains are included in this list, and other species as Paenibacillus polymixa has been studied

- Which sterilization method did you apply to obtain the sterilized low nutrient soil Inaho-baido? Could you explain the reason to use this method and why not use a method based in the growth of the seed directly on isolation soil? This procedure is alterating the conditions of soils (aireation, texture) and the potential populations. Also in the first inoculation of sterilized soil, did you differenciate between bulk soil and rhizospheric soil?

- The isolation method does not allow to differenciate between root endophytes and epiphytic bacteria

- Which is the criteria to select identified strain between 386 isolated microorganisms? This results are not included in the manuscript

- Which is the criteria to select identified strain between 386 isolated microorganisms? This results are not included in the manuscript

- Table 1. Identity percentage with 2 decimals

- Improve the quality of the figure 3

-It is not clear the relationship between the study of metabolism profile of strains performed with BIOLOG and the main objective of the work. Please clarify this part of the manuscript explaining which are the scientific reason to include this information in the manuscript.

-On subheading 3.4, appears (another time) the 386 isolated microorganisms, but quimiogenotipic analysis are performed with a limit number of microorganisms.

- Statical differences are respecto to positive control o negative control? this is a important issue, also treatments should be analyzed respect to  both controls and apply a robust statical analysis as ANOVA and Tukey test

Author Response

Thank you for reviewing our manuscript. Regarding the comments and recommendations, the manuscript was revised accordingly to your recommendations to some extent, please see the new version of the manuscript.

Round 2

Reviewer 2 Report

Authors made and extensive improvement in the revised manuscript, and it has significantly improved the quality of manuscript and scientific soundness. Although still manuscript has scope for English and grammar improvement.

Figure quality must be improved.

Reference formatting should be checked.

Author Response

Dear reviewer, 
Thank you for checking our manuscript. Regarding the comments and recommendations, the manuscript was revised. Last, the English in the manuscript was checked by a native colleague. 

Reviewer 3 Report

Dear authors,

I am glad to see that the manuscript has been improved by following the instructions properly. However, to complete the research line and the information I recommend including the results to test the isolated strains on plants that employ to select them. This information could be added as supplementary material.

After including this information, the manuscript can be accepted for publication.

Best regards

Author Response

Dear reviewer, 
Thank you for reviewing our manuscript. Regarding the comments and recommendations, the manuscript was revised. 

About the point of extra data as supplementary, all authors consider that the non-submitted data is too large, and all results and conclusions are based on the presented data. In addition, we did include the summary data in the respective tables and the description of the selection after revision. Also, the original data is kept in the original language (Japanese), which will take a long time to revise and adjust the journal format. If you are still interested, please get in touch with us by email.